# Improving Environmental Water Supply in Wetlands through Optimal Cropping Patterns

**Mahdi Sedighkia** [1,*] **and Bithin Datta** [2]

1   ICEDS & MSI, Australian National University, Canberra 2601, Australia
2   College of Science and Engineering, James Cook University, Townsville 4814, Australia
*   Correspondence: mahdi.sedighkia@anu.edu.au

**Abstract:** This study improves the environmental water supply in a wetland using a novel framework in which the environmental impacts due to irrigation supply and the economic losses for agriculture are minimized through the proposal of an optimal cropping pattern that changes the total cropping area and cultivated area of each crop. The ecological degradation functions for rivers and wetlands were developed using a fuzzy approach and data-driven model. The net farming revenue was considered as the economic index to maximize benefits. The root mean square error (RMSE) and the Nash–Sutcliffe model efficiency coefficient (NSE) were applied to evaluate ecological models. According to the results, the optimal cropping pattern simultaneously minimizes environmental impacts due to irrigation supply and maximizes farmers' benefits. The optimal cropping pattern provides more than 50% of the ideal net revenue on the catchment scale, which means that ecological degradations due to reductions in inflow in rivers and wetlands, as well as farmers' revenue losses, are minimized simultaneously. Furthermore, the results indicate that cropping patterns should be dynamic, which means that changing the cropping pattern annually based on the available water is essential to mitigating ecological impacts. This study demonstrates that the linking of cropping pattern optimization and environmental flow simulation in freshwater bodies should be considered in land-use policies due to the impact of cropping patterns on environmental degradation in wetland catchments.

**Keywords:** wetland ecosystems; economic losses; optimal agricultural land use; physical habitat simulation; data-driven model

## 1. Introduction

Sustainable development in agriculture has been highlighted in recent decades [1]. Unsustainable development in agriculture might affect different aspects of the environment. Irrigation is one of the requirements for farming in arid and semi-arid regions. The irrigation demand may be met using surface water resources or groundwater resources. Rivers are the main water resource for irrigating farms in many countries in the world. Increasing populations may be a serious threat to river ecosystems around the world [2]. Hence, the concept of environmental flow has been proposed in order to protect dependent species in rivers. Several methods can be used to assess environmental flow in rivers, including simpler methods, such as historic flow methods, and complex methods, such as habitat simulation methods and holistic methods [3,4]. Generally, advanced methods, such as ecological-based simulations, and holistic methods highlight regional ecological values, which means that these methods are more reliable for practical projects. However, some advanced methods might be expensive to implement in case studies. Instream flow incremental methodology (IFIM) was introduced for assessing physical habitats to determine ecological flow. In other words, physical habitat requirements can be applied to analyze environmental flow time series [5]. PHABSIM is a known software that applies univariate habitat suitability criteria to simulate a primary physical habitat [6]. As a brief

description of the univariate method, the suitability of different physical factors, such as depth, is computed in the first step. Then, the combined suitability can be calculated by combining the suitability of each factor using the geometric mean or other similar indices. This classic method has been criticized in the literature due to its weaknesses in terms of the interactions among physical factors. More details regarding the concept of interactions among parameters have been addressed in the literature [7]. Thus, multivariate methods have been proposed in this regard. One of the applicable methods is a multivariate fuzzy approach proposed for considering complexities of habitat selection with the development of verbal fuzzy rules [8]. The use of expert opinions to develop fuzzy rules is one of the important advantages of the fuzzy approach. Due to complexities in ecological assessment, the application of expert opinions might be effective in developing a reliable ecological model. This method has recently been applied as an independent method to assess and optimize an environmental flow regime [9]. Wetlands are another important inland water source that might need to be considered for the allocation of environmental water. Some studies have highlighted the environmental water requirements of wetlands [10]. It should be noted that considering integrated frameworks to assess environmental flow in river–wetland ecosystems is essential.

As presented, the role of irrigation is principal in developing farms in arid and semi-arid regions. Thus, it is important to use water in agriculture optimally. Different strategies have been used for optimal irrigation in the literature. One of the known strategies to reduce water demand is optimizing cropping patterns. According to the literature, two types of objective functions might be defined in which water consumption is minimized or revenue is maximized [11]. Linear and non-linear programming (LP and NLP) are the best-known methods for optimizing cropping patterns [12,13]. In fact, LP is the simplest method for optimization problems that can handle the linear objective function. In agricultural engineering, the linear objective function might be applicable in the simple forms of optimization problems. However, the complex design of a cropping pattern needs a non-linear objective function, which means that the development of an improved cropping pattern optimization model that considers environmental challenges, using advanced optimization algorithms, is essential.

The importance of environmental flow supply and the economic benefits of agriculture demonstrate that sustainable development in agriculture in arid or semi-arid regions needs an integrated optimization system in which the ecological impacts and farmers' revenue should be considered. This issue might be more complex in the basins that include wetland ecosystems. In other words, the ecological assessment of a river–wetland ecosystem is more complex compared with river ecosystems. Optimization might be a requirement for developing an integrated framework in this regard. Some simple methods, such as LP, might not be applicable to the complex problems of water resources or environmental management. The nature of many optimization problems is non-linear. In contrast, non-linear programming methods have been proposed to improve the solutions of non-linear functions applied in different disciplines [14]. It should be noted that the efficiency and the accuracy of the optimization algorithm should be observed simultaneously. Hence, novel methods, such as evolutionary algorithms, have been utilized extensively in recent decades. A wide range of algorithms, including new-generation methods, is usable [15]. Reservoir operation optimization is one of the known problems in water resource management that has been highlighted in the literature [14]. Different evolutionary algorithms have been utilized for the management of reservoirs or diversion dams [16]. It seems that evolutionary algorithms could be applicable to complex environmental and agricultural engineering problems.

According to the literature, the previous optimization models of cropping patterns have not considered the ecological impacts of water supply on inland water bodies in the structure of the model. It should be noted that using surface water to irrigate lands would reduce the instream flow in the water bodies, including rivers and wetlands, significantly. Hence, developing novel cropping pattern models in which the environmental challenges

of irrigation supply by freshwater bodies including wetlands and rivers are integrated into the optimization system is a research gap and is the main motivation behind this study. This study proposes a novel combination of ecological models and an economic cropping pattern model to optimize the total cropping area and cultivated area of each crop. The proposed method might be helpful in mitigating the ecological impacts of water supply at the river basin scale considerably. In other words, it can add the environmental degradation of wetlands and rivers to agricultural land-use policies.

## 2. Study Area

Urmia Lake located in the northeast of Iran is one of the largest saltwater lakes in the world, which has suffered from inadequate ecological water levels in recent decades due to increasing cultivated areas in the catchment and consequently agricultural water demand. Hence, improving environmental water demand is necessary in this catchment which might be possible using changing the cropping pattern as one of the solutions. In other words, an optimal cropping pattern might help to improve environmental water by minimizing the impacts on the net revenue.

According to the initial assessment, eight rivers are the main inflows of the wetland and evaporation is the main outflow for this lake. Figure 1 displays the catchment of Urmia Lake. These rivers are responsible for supplying the available water in the lake and were considered in the simulation of habitat suitability in which hydrometric stations were available upstream and downstream. Based on the developed methodology, a fish species (*Capoeata capoeta*) was selected as the target species for the physical habitat simulation in the rivers. Moreover, *Phoenicopterus* was selected as the target bird species for evaluating the habitat suitability of the lake based on a previous study [17]. Evaporation from the surface of the lake is one of the components in the optimization model, which was considered based on the recorded data from the local weather station.

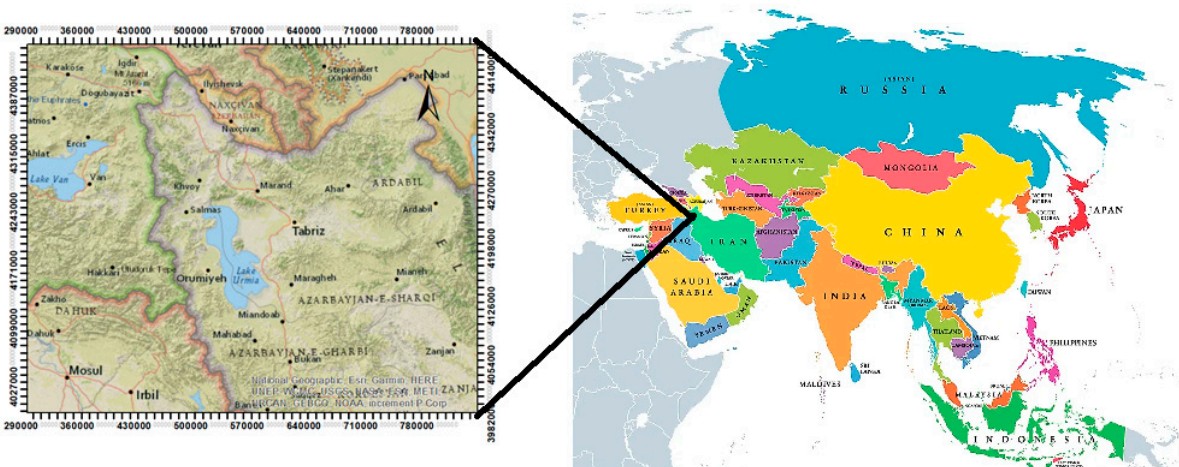

**Figure 1.** Location of Urmia Lake.

The present study highlights cultivation patterns. Thus, it is required to present more details regarding the selected crops in the case study. We selected some main crops in the present study based on the recommendations of the regional agricultural authority. These crops are the most popular crops for the farmers in the study area including alfalfa, almond, apple, barley, grape, potato, sugar beet, and wheat. More information regarding crops is needed to develop the cultivation pattern model including the cost of cultivation, price, irrigation demand, and yield. Figure 2 displays the cost of the cultivation and the price of the selected crops. Moreover, Figure 3 displays the yield and the irrigation demand for the crops. Table 1 shows all the basic information about the study area, which can be helpful for a better understanding of the purposes of this research work.

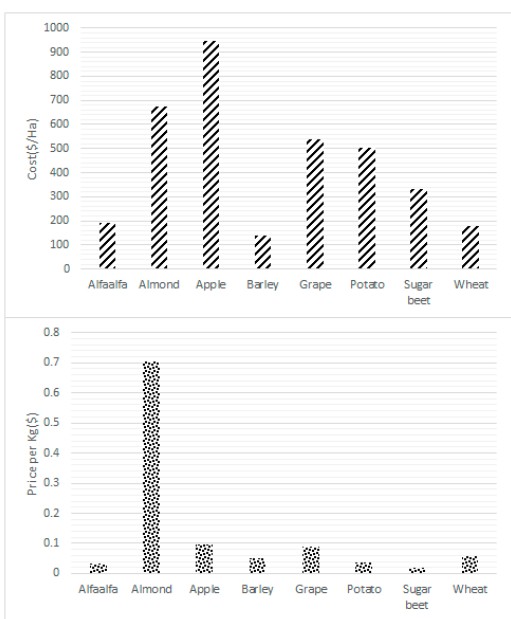

**Figure 2.** Cost of cultivation and price of the selected crops in the optimization model based on regional costs and prices.

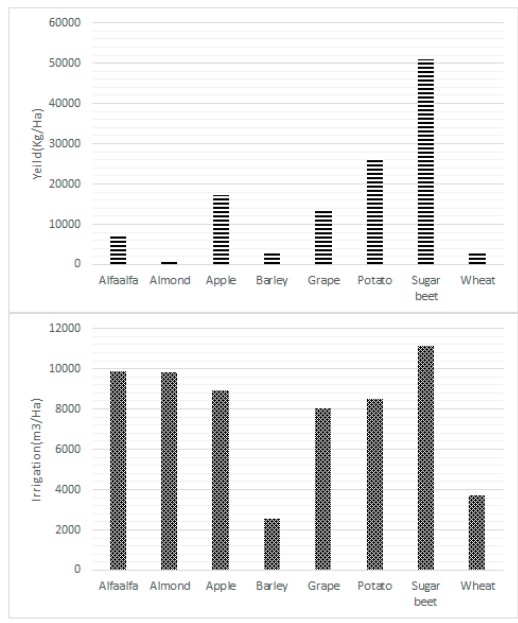

**Figure 3.** Irrigation demand and yield of the selected crops in the optimization model.

**Table 1.** More details regarding the features of the study area.

| Feature | Description |
|---|---|
| Usual crops in the wetland basin | Alfalfa, almond, apple, barley, grape, potato, sugar beet, wheat |
| Sources of inflow to the wetland | Main rivers are the sources of the inflow to the wetland considered in the ecological simulation of this study |
| Catchment area of the wetland | 51,876 km$^2$ |
| Area of wetland (Urmia Lake) | Original area 5200 km$^2$ (However, part of it is dried due to a lack of enough environmental water) |

## 3. Methodology

### 3.1. Overview

It is essential to have an overview of the proposed optimization model. The proposed framework includes two ecological models and one economic model. The first ecological model reduces destructive impacts on the river habitats. Moreover, the second model mitigates impacts on the wetland ecosystem. The economic model maximizes the net revenue of farming by changing the cropping pattern at the catchment scale. In other words, we applied two ecological models to the structure of the agricultural land use optimization. Figure 4 displays a flowchart of the proposed methods. A four-year period was selected for implementing the optimization model to obtain the optimal cropping pattern from 2010 to 2014. Details of the simulations and the optimization will be provided in the following sections.

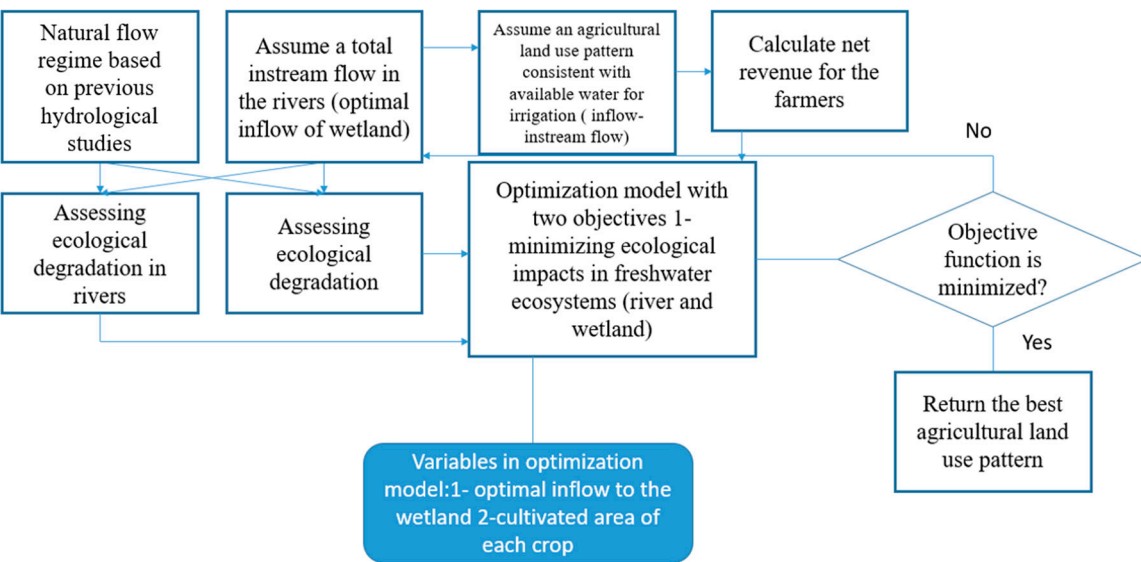

**Figure 4.** Flowchart of the proposed method.

### 3.2. Ecological Degradation Assessment (Rivers)

We aimed to minimize the ecological impacts on the river ecosystems as well as the wetland ecosystems. As the inflow of the wetland, these rivers are important habitats for fish species, which means habitat suitability in the rivers might be very important to mitigate environmental impacts. Thus, we selected an aquatic species as the target to simulate environmental suitability in the rivers. Assessing habitat requirements in the rivers is possible using different methods. According to the literature, physical habitat suitability is an important aspect in the assessment of aquatic habitats [18]. In fact, an appropriate combination of physical parameters should be available in the assessment of the habitats. We focused on physical habitat loss in this study to mitigate environmental losses for the aquatic habitats in the rivers as the inflow of the wetland ecosystem. A fuzzy approach was applied to simulate the river habitats due to the presented advantages in the Introduction. Figure 5 displays the workflow of habitat modeling in rivers. A weighted useable area–discharge relationship was generated as the output of the habitat modeling in all rivers. We selected a representative reach downstream of the rivers to simulate physical habitats. If adequate environmental flow is available downstream of the rivers, it will mean all fish habitats from upstream to downstream would have adequate instream flow. Moreover, available flow downstream (after the selected river reaches) of the rivers will drain to the lake because no more water abstraction is available downstream. In fact, several reservoirs or water abstraction projects have been built at the midstream of rivers that are responsible for conveying water to farms. Thus, selecting the representative reach downstream of the rivers is logical to minimize the ecological impact on fish habitats.

More information regarding ecological modeling in rivers can be helpful for the readers. The target species was selected based on long-term fish observations related to several studies in this region and initial ecological studies in the study area. In fact, the initial studies indicated that this species can be observed in all rivers in the wetland basins, which means it can be a target species for all physical habitat simulations. Moreover, its population is sensitive to the physical flow parameters, which means it can indicate the impact of changing physical habitats for assessing the ecological flow regime. The fish observations were carried out with the electrofishing method, which is one of the known methods for ecological observations in river habitats. Then, the results of the fish observations were discussed in an expert panel including two regional ecologists and our research team to convert the direct observations as well as other long-term observations into the fuzzy rules. The fish observations were carried out in more than 100 points in the winter and summer of 2017 in which depth, velocity, and bed particle size were measured simultaneously. In the hydraulic simulation, a DEM file with a 10 m resolution was applied. Furthermore, we defined three membership functions including low, medium, and high considering three physical parameters including depth, velocity, and channel index (bed particle size), which means 27 fuzzy rules from the expert opinions were developed for physical habitat simulation.

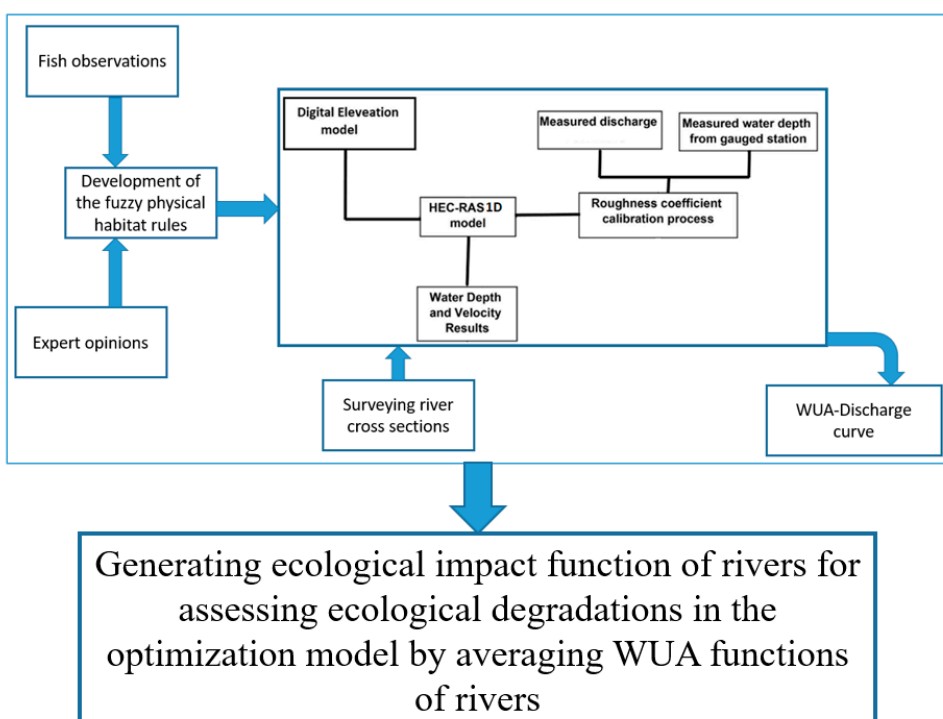

**Figure 5.** Habitat modeling framework in rivers.

### 3.3. Ecological Degradation Assessment (Wetland)

Different bird and fish species need wetland ecosystems for biological activities. In other words, these species might use the environment of the wetland as part of their lifecycles. Thus, the suitability of the wetland ecosystem matters as well. Considering all available species in this assessment is not possible in practice. Thus, using one target species might be reasonable in the rivers as well as the wetland. The selected target species should have a strong relationship with the ecological status of the ecosystem. Based on previous research works on this case study, the flamingo is an important bird species in our case study. These birds are immigrants that might spend some months in Urmia Lake. The main biological activities of this bird such as reproduction need suitable wetland habitats. We developed a data-driven model in this regard. Artificial neural networks are one of the

known methods to develop a robust data-driven model in ecology [19]. However, due to the requirements of improving neural networks, these models have been improved [20]. Adaptive neuro-fuzzy inference systems improve the interpretability and robustness of outputs by combining a neural network and a fuzzy inference system [21]. We developed an ANFIS model to simulate Urmia Lake habitats considering the selected bird species. Figure 6 shows the architecture of the ANFIS for assessing the wetland ecosystem.

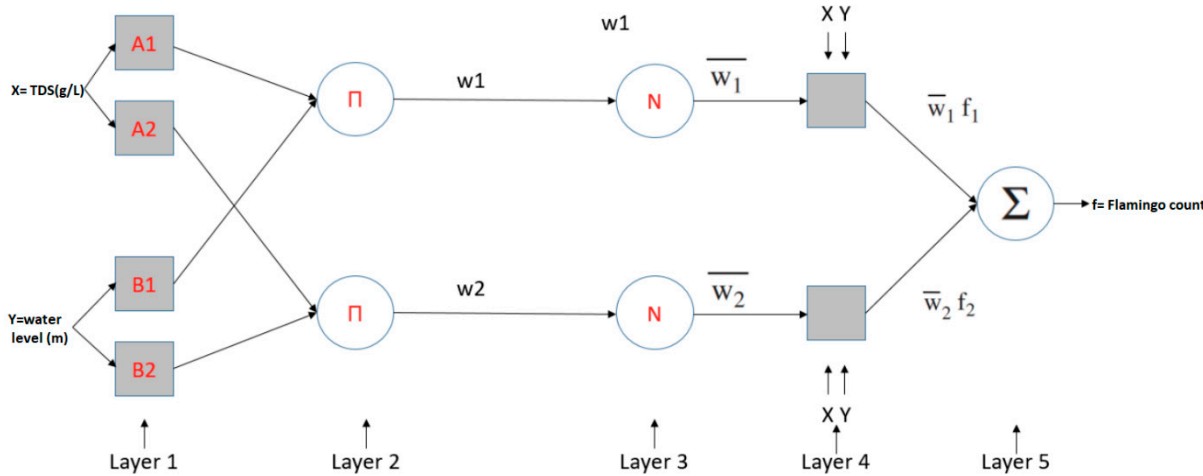

**Figure 6.** General architecture of an ANFIS with two inputs.

Many parameters might be effective in the population assessment of the flamingo in the wetland. However, according to previous studies in this case study, we selected two key parameters for the data-driven model including total dissolved solids (TDS) and water level of the wetland [17]. It should be noted that other physical and chemical parameters might affect these parameters. Hence, selecting these two parameters seems logical in the developed framework. Table 2 shows more details of the ecological model in the wetland. The Nash–Sutcliffe model efficiency coefficient (NSE) and root mean square error (RMSE) were utilized to measure the robustness of the data-driven model and physical habitat model as follows.

$$\text{NSE} = 1 - \frac{\sum_{t=1}^{T}(M_t - O_t)^2}{\sum_{t=1}^{T}(M_t - O_m)^2} \tag{1}$$

$$\text{RMSE} = \sqrt{\frac{\sum_{t=1}^{T}(M_t - O_t)^2}{T}} \tag{2}$$

where $M_t$ is a simulated value, $O_t$ is an observed value, and $O_m$ is the mean of the observed values in the samples, and T is the total number of samples. Many previous studies have been carried out for a better understanding of the ecological challenges in Urmia Lake [17]. All these studies highlighted that the flamingo is one of the terrestrial species that is highly dependent on the water level and total dissolved solids (TDS). In other words, changing either the water level or TDS can change the flamingo population considerably. Moreover, the flamingo is the only major terrestrial species that stays in the lake habitats for a long time during immigration. Hence, all these environmental considerations justify selecting it as the environmental indicator for the wetland. Flamingo observations have been carried out by the regional environmental department in the recent decade (in different years of the recent decade), which were applied in this study for developing a data-driven model. Based on comparing the results of the model and observations considering Equations (1) and (2) as the evaluation indices, the flamingo model was validated. The results will be presented in the result and discussion.

**Table 2.** More details of the ecological model for the wetland ecosystem.

| Inputs | Membership Function Number (Inputs) | Membership Function Type (Inputs) | Output | Membership Function Number (Output) | Membership Function Type (Output) | Clustering Method |
|---|---|---|---|---|---|---|
| Water level in the lake from the sea (m) | 10 | Gaussian | Flamingo count | 10 | Linear | Subtractive clustering |
| TDS (g/L) | 10 | Gaussian | | | | |

### 3.4. Optimization Model

Equation (3) shows the proposed objective function in which DRV is the difference between optimal revenue and ideal revenue of farming in the catchment of Urmia Lake. Moreover, DFC is the difference between flamingo count in the optimal environmental water and natural flow. Similarly, DWUA is the difference between the weighted usable area in the optimal environmental flow and the natural flow in the rivers.

$$Minimize(OF) = (DRV)^2 + (DFC)^2 + (DWUA)^2 \tag{3}$$

Defining some constraints is necessary in the proposed optimization model as follows:

1.  The total area of the cultivated land should not be more than the available land in the study area
2.  Allocated water to the farms and orchards should be the same as the release for agriculture in the rivers.

The penalty function method was applied to add constraints in the optimization model [22]. Furthermore, we utilized invasive weed optimization (IWO) to optimize agricultural land use in the present study [23]. Figures 7 and 8 show a flowchart of the optimization model and IWO, respectively. It should be noted that we applied RMSE and NSE for evaluating the optimization model in terms of physical habitat loss, in which physical habitat suitability in the natural flow and the optimal environmental flow were considered as the observation and simulation, respectively.

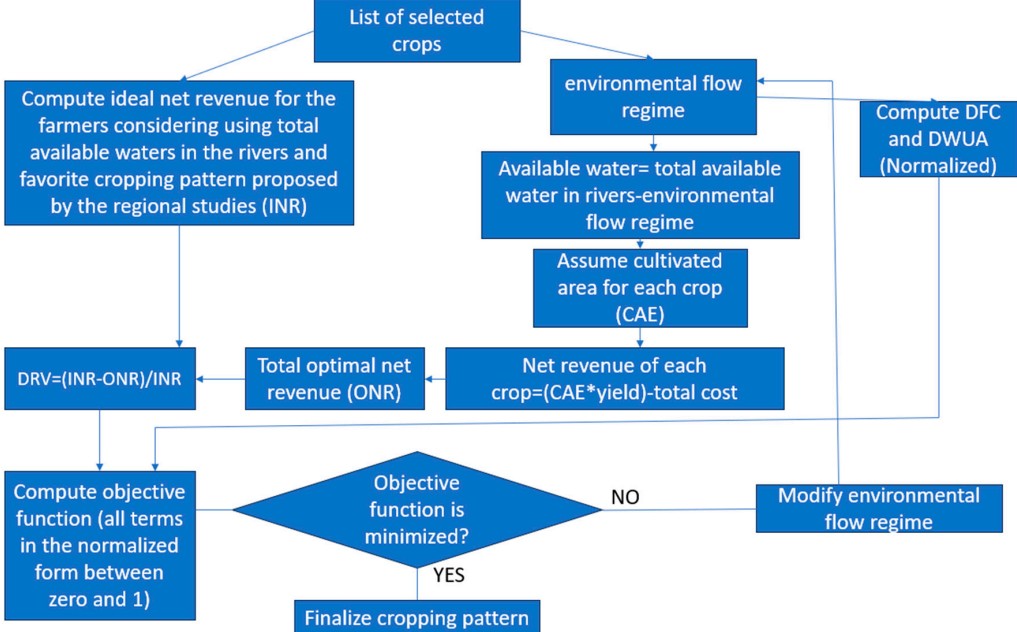

**Figure 7.** Flowchart of the cropping pattern optimization.

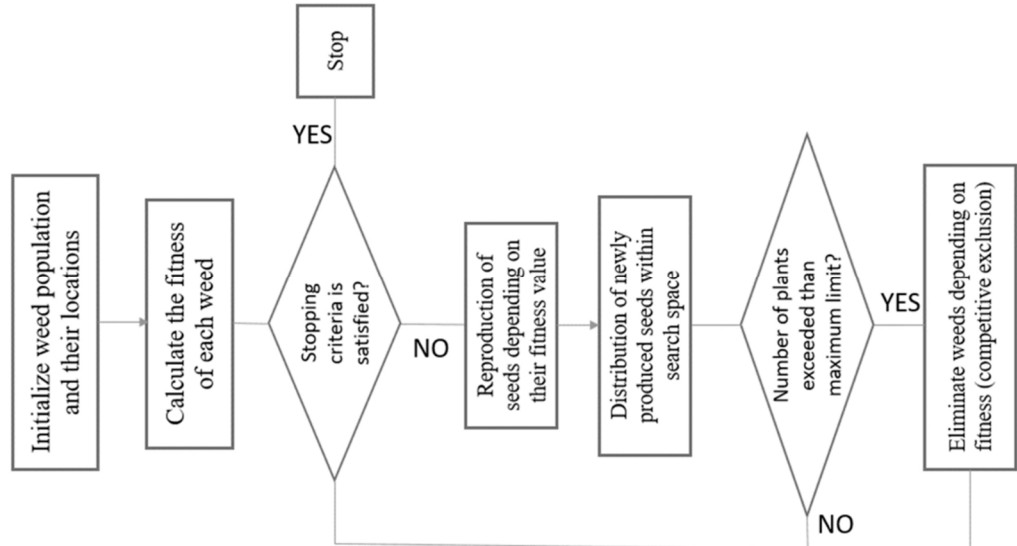

**Figure 8.** Flowchart of invasive weed optimization.

It is needed to explain more details regarding the optimization model and how the objective function can improve environmental water requirements as well as maximize benefits for the farmers. The tree available terms (DRV, DFC, and DWUA) are dimensionless, which means all of them would change between zero and one as the normalized terms. DRV is the difference between ideal net revenue and optimal net revenue to ideal net revenue. Furthermore, DFC is the difference between the flamingo count in the natural flow and optimal environmental flow to the flamingo count in the natural flow. Similarly, DWUA is the difference between the weighted usable area (WUA) in the natural flow and the optimal environmental flow to the WUA in the natural flow. In fact, DRV is a term for minimizing the loss of the benefits for the farmers, and two terms including DFC and DWUA can improve environmental water by minimizing the loss of fish habitat suitability as well as the flamingo population. In other words, the proposed objective function can minimize losses in the farmer community as well as habitat losses simultaneously, in which minimizing habitat loss means improving environmental water requirements.

## 4. Results and Discussion

As a first step, it is essential to present the result of the physical habitat simulation in the rivers and the training and testing process of the data-driven model for the wetland ecosystem. Figure 9 displays the normalized average weighted useable function (ecological impact function) for the rivers as the inflow of Urmia Lake. It seems that the ecological response of the rivers is different. Figure 10 displays the results of the testing process for the ANFIS-based model of the wetland ecosystem. We applied 80% of the available data for the training process and the rest for the testing process. Two measurement indices including NSE and RMSE were utilized to measure the robustness of the data-driven model. As displayed in the figure, NSE and RMSE are 0.98 and 642, respectively, which indicate the robustness of the model to simulate the flamingo population. We simulated the average population of the flamingo in the spring. Flamingos immigrate to Urmia Lake in this season for reproduction and other relevant biological activities. Their population in the spring is strongly dependent on the water level and TDS. Thus, we simulated the average flamingo count in the spring in each year of the simulated period. According to the literature, if NSE is more than 0.5, it indicates the predictive skills of the model are robust and reliable for future applications. Moreover, the RMSE demonstrates that the model does not have considerable error in simulating the flamingo population. It is too low compared with the maximum observed population of flamingos in the study area.

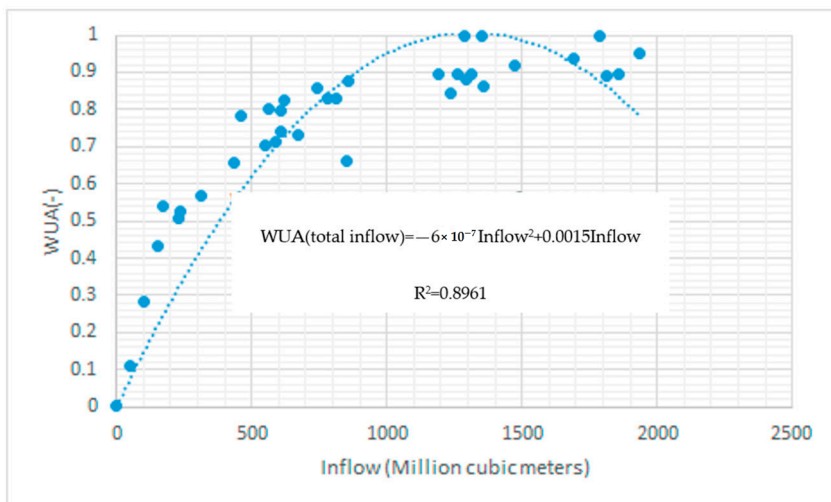

**Figure 9.** Output of fuzzy physical habitat simulation in the rivers (ecological impact function used in the optimization model).

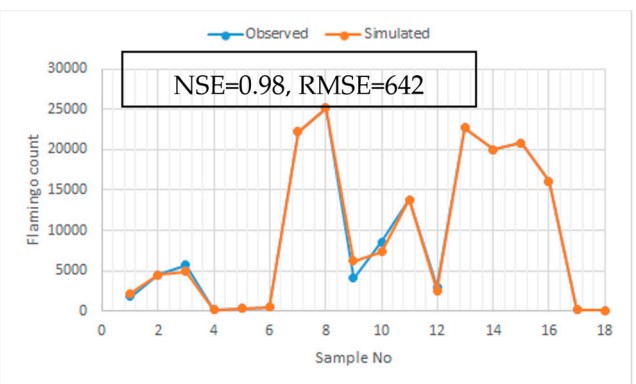

**Figure 10.** Testing process of the data-driven model.

As a next step, it is necessary to present the results of the optimization model. As presented, three main purposes were defined for the model including mitigating the ecological impacts in the rivers and in the wetland and maximizing the agricultural land use with the development of optimal agricultural land use in the study area. Thus, we present the results in these parts respectively. Figure 11 displays NWUA in the natural total inflow and in the optimal total instream flow in the rivers. Based on the measurement indices of the optimization model including NSE and RMSE, the performance of the optimization model is highly robust because it can minimize the difference between habitat suitability in the natural flow and the optimal instream flow (RMSE = 0.16, NSE = 0.75).

Figure 12 displays the natural flow, optimal instream flow, and available water for farming in the study area. Figure 13 displays the results of optimization in the wetland ecosystem. It shows that the optimization model can balance the needs of the environment and agriculture fairly. The water level proposed by the optimization model in the lake is not far from the level of the lake in the natural flow. Thus, the performance of the optimization model is defensible in terms of water level. However, the main criterion is the flamingo population in the wetland ecosystem assessment. In other words, it is important to show how optimization can protect the flamingo population in the wetland ecosystem. Figure 13 displays the flamingo count in the natural condition and optimal condition as well. It should be noted that we simulated a challenging period in the wetland ecosystem. The flamingo population, even in the natural conditions, is too low in the third and fourth years of the simulated period. The population of the flamingos in the third and fourth years is low in both conditions. The average population of the flamingos by the optimization model

and in the natural flow demonstrates that the performance of the optimization model is acceptable and robust. Another purpose of the optimization model is to maximize the net revenue of the farmers by proposing the optimal agricultural land use in the basin. In fact, models optimize the cultivation pattern dynamically in each year of the simulated period.

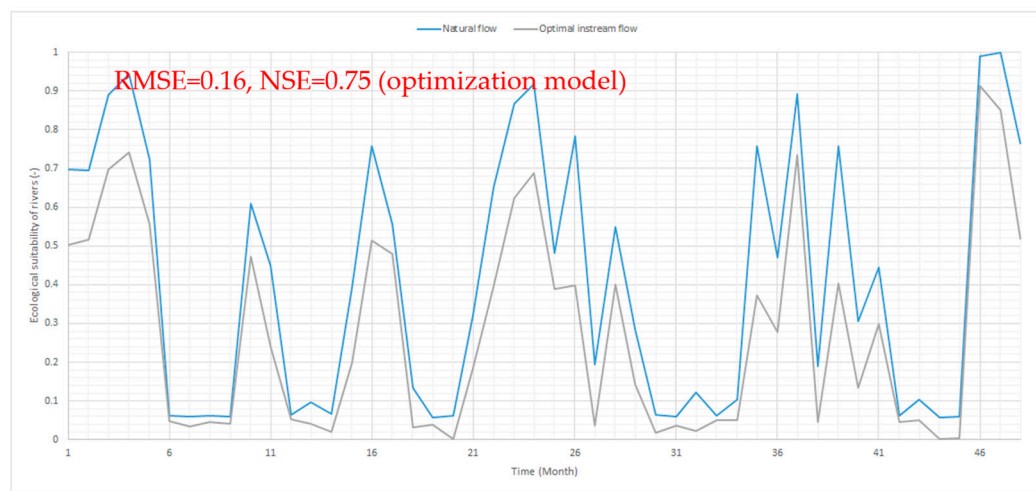

**Figure 11.** WUA in the simulated period for all the rivers connected to the wetland.

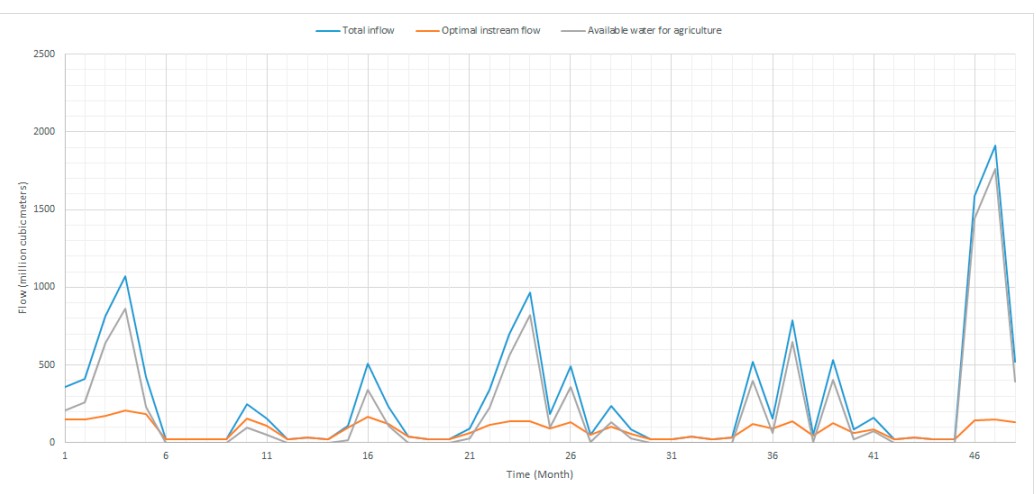

**Figure 12.** Release for the environment from different rivers with the inflow of the wetland.

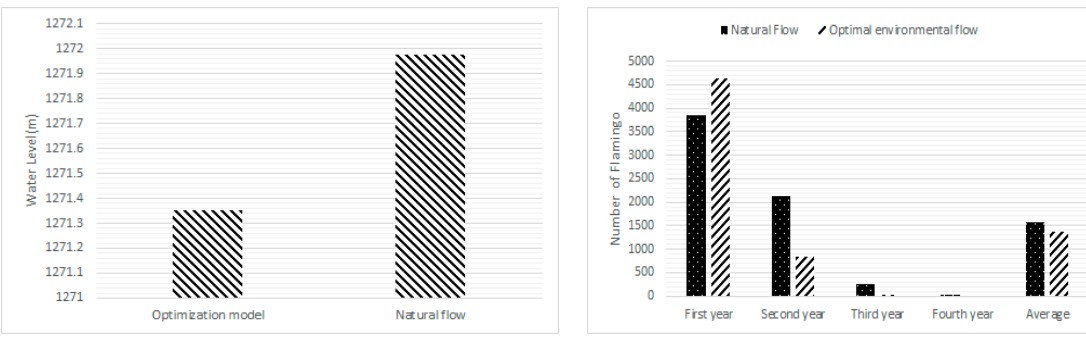

**Figure 13.** Results of the optimization model in the wetland ecosystem.

Figure 14 displays the optimal agricultural land area and net revenue for the farmers proposed with the simulation–optimization system. It should be noted that the maximum

land area and ideal net revenue were considered at 295,000 Ha and USD 130 M based on the recommendations by the regional agriculture authority. The ideal revenue has been assessed based on the cultivation in all the areas by the current agricultural land use. It seems that cultivated area should dramatically be decreased in the optimal condition compared with the maximum available area. However, the optimization model can protect 65% of the net revenue in the best condition. Figure 15 displays the optimal cultivation pattern in different years of the simulated period. The results demonstrate how the dynamic ecological condition of the ecosystem could be effective in the planning of agriculture in the basin. For example, the cultivated area of the sugar beet changes in the simulated period in the range of 11% to 52%, which is a broad range. It seems we face a complex problem to design the optimal pattern of agricultural land use. In other words, agricultural land use should be dynamically designed in terms of cultivated crops consistent with the ecological impacts on inland water bodies.

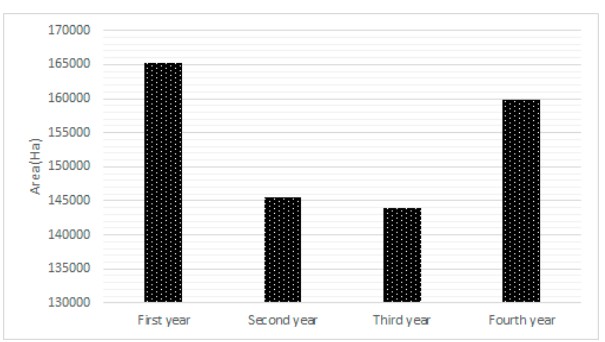
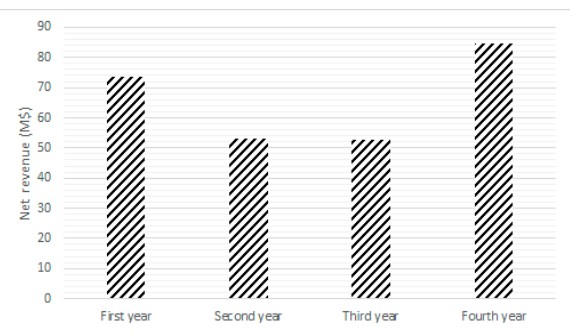

**Figure 14.** Optimal area of the agricultural lands and net revenue proposed using the simulation–optimization model.

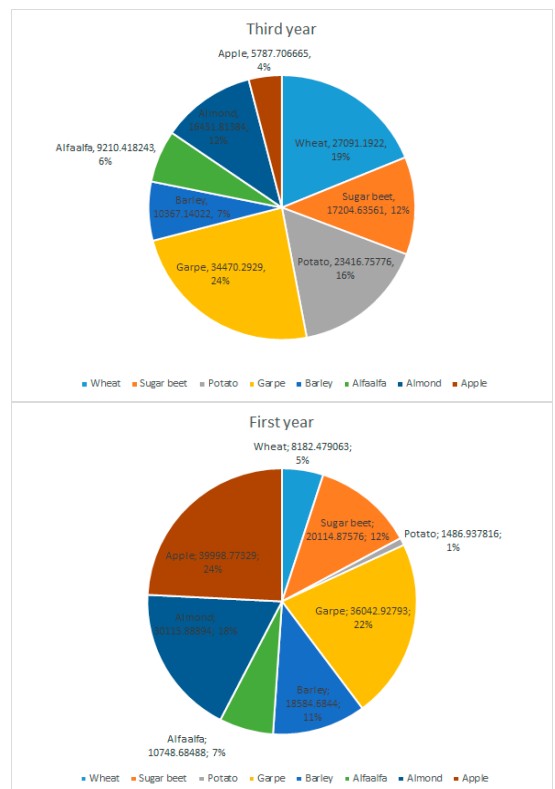
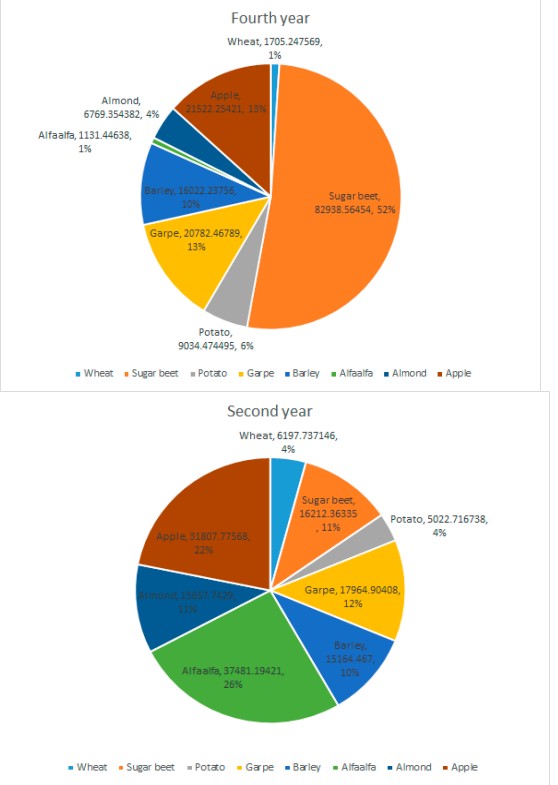

**Figure 15.** Optimal agricultural land use in different years of the simulated period proposed using the simulation–optimization model.

A full discussion is essential regarding the outputs of the present study. In fact, each optimization system might have strengths, drawbacks, and limitations that should be noted in the applications. Moreover, it is required to compare the result of the present study with previous studies regarding the optimization of the cropping pattern. Previous studies highlighted the importance of coupling ecological models and economic models in the optimization of agricultural land use [24]. However, they considered other aspects of ecological modeling such as minimizing the use of chemical fertilizers and chemical pesticides. It seems that the proposed method considers a new and important aspect in the ecology of the structure of agricultural land use optimization. In fact, we assessed the ecological impacts of reducing instream flow in the rivers and available water in the wetland ecosystem. We recommend considering unstudied ecological impacts in future studies, which is a requirement for the sustainable development of agriculture in the basins. As presented in the introduction, LP methods have been applied for the optimization of cropping patterns extensively [12]. However, the developed objective function in the present study indicates that LP methods are not applicable to the complex system of cropping optimization. In other words, when ecological and economic models are combined to optimize agricultural land use, LP methods are not able to provide an optimal solution for the pattern [25]. The present study demonstrates that adding robust ecological views in the optimization model of agricultural land use is essential. Moreover, the ecological impacts should not be limited to the pollution due to agriculture in the water bodies. In fact, a complex relationship should be considered for linking the supply of water from inland water bodies such as rivers and wetlands and ecological impacts. Previous studies addressed all ecological complexities in the wetlands [26].

The proposed mechanism was successful in this case study, and it could be interesting to apply the method in future projects. The developed objective function could be used in case studies including river–wetland ecosystems. However, the proposed method is flexible, which means it is possible to utilize it for the river network as well. For example, the proposed method could be applied in the diversion dams or reservoirs that are responsible for irrigation supply in many regions. However, some modifications to the optimization system are needed. Hence, we recommend developing the proposed method for other applications and other types of water supply systems. In fact, we applied the proposed method in the watershed scale to design optimal agricultural land use. However, it is useable for smaller scales such as diversion dam projects. Previous optimization systems for agricultural land use are not applicable and efficient to face environmental challenges in the river basin and wetland ecosystems. It is required to link ecological models and economic models in an integrated system, which is the main advantage of the proposed method.

An in-depth discussion regarding the importance of the research gap is needed. Current methods for environmental flow assessment in rivers as well as wetlands are only able to assess required flow. However, the main challenge is identifying how this environmental flow can be supplied, especially in dry areas with considerable water consumption. Many areas such as our study area suffer from considerable agricultural water demand, which limits the available water for the environment. There is a serious need in water resources management to integrate environmental flow methods and optimization models of water supply. Furthermore, cropping patterns are highly effective on the water demand, which is not highlighted in previous environmental flow management studies. This issue is more important in wetlands that might have large catchments with many agricultural lands. In fact, environmental managers do not currently have robust tools to integrate cropping patterns and environmental flows in wetland catchments. This study links environmental flow and cropping patterns to provide the optimal scenarios of agricultural planning with a focus on cropping patterns in which environmental water requirement is improved.

The fuzzy approach was applied to simulate physical habitat suitability, which is advantageous for this study as well as future similar studies due to the following reasons. First, habitat selection by fish is a complex process in which some unknown parameters might be effective. Field studies and measurements cannot consider unknown parameters,



though qualitative observations by experts can integrate the impact of these unknown parameters. Hence, using a fuzzy approach that can use the results of field studies as well as expert opinions is highly advantageous to developing correct habitat suitability criteria. Moreover, field studies in fish habitats would generally be carried out in shorter periods, which means statistical approaches cannot consider observations from some years ago by local experts. However, the fuzzy approach can cover this deficit by considering expert opinions in fuzzy rules.

Computational complexities are one of the important aspects in the application of optimization systems in practical projects. According to the official definition, computational complexities are the time and memory required for the optimization algorithm to find the best solution [27]. High computational complexities might be a serious concern for using the optimization system in practical projects. It might be needed to have numerous simulations. Moreover, covering a long-term simulation period is another requirement in practical projects. This issue might be problematic, especially in terms of the running time of the optimization algorithm. Hence, the analysis of the proposed method in terms of computational complexities is one of the important aspects of the discussion. We applied WUA functions and a data-driven model of the flamingo population in the optimization model that is effective on the complexities. It should be noted that using WUA functions might be an advantage for the optimization system. They do not increase computational complexities. On the other hand, utilizing an ANFIS-based model in the structure of the optimization model would increase computational complexities, especially running time, considerably. In fact, it forces the optimization algorithm to open the ANFIS-based model repeatedly, which is time-consuming. Thus, we can claim that the computational complexity in the proposed model is not very high. However, it is not low due to using the ANFIS-based model to simulate the flamingo population in the wetland ecosystem. In other words, applying the fuzzy physical habitat simulation might be a strength, and using an ANFIS-based ecological model might be a weakness for the optimization system in terms of computational complexities. We recommend focusing on reducing computational complexities in future studies to increase the efficiency of the proposed method. For example, it might be possible to use other types of machine learning or deep learning methods to reduce computational complexities. More details of other options have been addressed in the literature [28].

Another issue that should be discussed is the efficiency of the optimization algorithms. We utilized IWO as the optimization algorithm in the present study. Using this evolutionary algorithm might have some advantages including higher efficiency and applicability compared with previous methods such as LP and NLP methods [29]. However, there is no guarantee for global optimization using evolutionary algorithms. Thus, many evolutionary algorithms have been developed in the literature. We only used IWO for the developed optimization system. It is recommendable to apply other types of evolutionary algorithms in this regard. For example, different new generation algorithms including animal and non-animal inspired algorithms could be used to improve the optimization solution. It is essential to compare the results of future studies with the present study to increase the applicability and efficiency of the developed model. Clearly, it could be observed that the developed objective function contains some terms. We aggregated these terms in one objective function. However, it is possible to apply multi-objective evolutionary algorithms such as the multi-objective genetic algorithm or multi-objective particle swarm optimization. However, we use the aggregated form due to some significant advantages. First, it is more efficient in terms of computational complexities. As discussed, the computational complexities in the current condition are not low. Using multi-objective optimization algorithms increases computational complexities remarkably. In fact, they utilize more complex algorithms to find the best solution for the objective functions. The result of the present study demonstrated that applying an aggregated form could generate the optimal solution. Thus, it might be an advantage for the developed model to use an aggregated form of the objective function. Furthermore, it should be noted that a limited number of multi-objective

functions have been developed in the literature. In fact, many new-generation algorithms are not available in the multi-objective form. As highlighted, one of the disadvantages of evolutionary algorithms is the lack of a guarantee for global optimization [30,31]. Thus, the need for applying different algorithms in practical projects might limit the application of multi-objective optimization algorithms.

It is required to discuss the shortcomings of the developed model. This issue might be helpful in terms of two aspects, including the future research needs and analysis of the risks of using the proposed model in practical projects. We considered the physical habitat suitability in the assessment of the ecological impacts on the rivers. It should be noted that physical habitat loss was the main problem in the rivers of this case study. Thus, we highlighted physical habitat loss in the present study. However, water quality impacts might be a problem in some cases as well. For example, reducing the instream flow might reduce the suitability of the water quality for aquatic species. Hence, we recommend adding the water quality simulation of the rivers in the structure of the optimization model to optimize agricultural land use in case studies in which water quality might be problematic for aquatic habitats in rivers. Moreover, we considered an ANFIS-based ecological model with a focus on flamingos to simulate habitat suitability in the wetland ecosystem. However, it might not be applicable in all case studies. In fact, selecting a target species for the simulation of habitat suitability in a wetland ecosystem might not be simple. In fact, an initial ecological assessment is needed to recognize target species. We utilized the results of the previous studies to select the target species. It is probable other aquatic or terrestrial species might be appropriate for simulating wetland ecosystem suitability in other cases. It should be noted that it is essential to consider the requirements of each case study for selecting the target species. The selected target species should be strongly sensitive to the wetland water level.

Selecting crops might be another challenge for practical applications of the developed model. It should be based on technical and social considerations. A social assessment might need to be added to the developed model in future studies. Moreover, the suitability of the soil might be another problem for designing the cropping pattern. It is recommended to add a suitability assessment of the farms to the developed model in future studies. Another advantage of the proposed model is that it can minimize negotiations between stakeholders and environmental managers. It should be noted that release for the environment is a controversial issue in the management of basins. Thus, an optimization model is required to reduce negotiations. The flexibility of the proposed model should be highlighted, which means other types of environmental models or economic models for agriculture could be used in the structure of the optimization model. Supply of urban water demand might be important in some cases. However, it was not a major issue in our case study. Thus, adding the urban water demand function to the optimization model might be another improvement in future studies.

The spatial pattern of crops in the basin should be discussed due to its importance, as highlighted in the literature [32]. In this study, we proposed optimal cropping patterns in the basin that can be used in future agricultural planning to improve the environmental water requirement of the wetland. Another important point that should be noted in future studies of Urmia Lake is the spatial pattern of the proposed optimal cropping pattern. The current spatial pattern of the farms can be a guideline to obtain the spatial pattern of the optimal cropping pattern because the impact of other factors such as soil has been considered. In other words, the optimal pattern can be implemented based on current directions in the spatial patterns. For example, the area of alfalfa can be changed by farmers' contribution to finding the best lands in this regard. However, it is recommended to carry out a further spatial study to find the best spatial pattern of the proposed optimal cropping pattern in which the impact of other effective parameters such as soil features and agricultural practice can be considered.

To sum up, the present study demonstrated that the cropping pattern is highly effective on the ecological degradations of inland water bodies such as rivers and wetlands which

implies a land use policy should be implemented consistent with the ecological impacts on the water resources. The absence of a linkage between cropping patterns and ecological impacts of water bodies is highly destructive in terms of environmental degradations in the river basin. The present study proposed an optimization model for the dynamic design of cropping patterns in terms of total cultivated area and the area of each crop. We recommend using the proposed system to design optimal cropping patterns at the river basin scale and correcting agricultural cropping patterns consistent with the outputs of the optimization model.

## 5. Conclusions

This study evaluated a novel cropping pattern optimization model considering ecological degradations in the freshwater ecosystem and farmers' net revenue. In the case study, the ecological degradations in the rivers and the wetland could be mitigated by proposing optimal agricultural land use, while the net revenue of farming was maximized. Using fuzzy approaches and data-driven models for simulating ecological degradations linked with the optimization model could be an effective solution for modeling the cropping pattern. This study demonstrated that combining cropping pattern models and hydro-ecological models of water bodies should be considered in land use policies due to the impact of cropping patterns on environmental degradation in wetland ecosystems.

**Author Contributions:** Conceptualization, M.S. and B.D.; methodology, M.S.; software, M.S.; formal analysis, M.S.; data curation, M.S.; writing—original draft preparation, M.S.; writing—review and editing, B.D.; supervision, B.D. All authors have read and agreed to the published version of this manuscript.

**Funding:** This research received no external funding.

**Institutional Review Board Statement:** Not applicable.

**Data Availability Statement:** Some or all data and materials that support the findings of this study are available from the corresponding author upon reasonable request. However, it is not free of charge.

**Conflicts of Interest:** The authors declare no conflict of interest.

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
