# Peer review of "Improving Environmental Water Supply in Wetlands through Optimal Cropping Patterns"

_agriculture, doi:10.3390/agriculture13101942_

Round 1

Reviewer 1 Report

The authors present an interesting study to combine ecological models and economic cropping pattern models to optimize cropping area and cultivated area for different crops. One of the principal arguments for their objectives is that optimizing the cropping pattern will help mitigate the ecological impacts of water supply in the river basin.

My principal concern about the paper is the methodology followed by the authors; there is a lot of information missing about how the data for the study was obtained, the study period, and the methods. I highly recommend that the authors pay attention to the following major and minor observations:

Major observations

In the sections "Ecological degradation assessment (rivers)" and "Ecological degradation assessment (wetland)". The use of the fish species must be justified. How were the fish observations, flamingo observations, and expert opinions obtained? How many fish observations and expert opinions were used to develop the fuzzy functions? How many flamingo observations were used to develop the ANFIS model? Why is the abundance of flamingos an indicator of ecological degradation? For both models, how were the data partitioned to validate the models? Which is the DEM resolution?

For the optimization model, How many hydrometric stations were used, and where are they located?

It is important to mention the dates on which the data were obtained (flamingo abundances, crops, fish habitats, flows, and water depth) and analyzed (simulated period).

Where are the fish in the results?

The discussion has a lot of affirmations without support; please support them. Please consider writing about the spatial pattern of the crops inside the basin.

Minor Observations

Use the appropriate citation and reference styles for the journal.

Line 123 specifies that the aquatic species is a fish.

Line 124 Put the scientific name If you do not know the specific species, just put Phoenicopterus sp.

Figure 1 and Figure 2 increase the size of each graph and put them on a horizontal

Figure 10: Sample No. of What? Is it a time associated with the sample No.?

Line 266, figure 8?? I think it's 11. WUA, not NWUA

Figure 11 I do not understand why the NSE is observed vs. the model?

Figures 11 and 12 Please use the same colors for the same variables.

Author Response

The authors present an interesting study to combine ecological models and economic cropping pattern models to optimize cropping area and cultivated area for different crops. One of the principal arguments for their objectives is that optimizing the cropping pattern will help mitigate the ecological impacts of water supply in the river basin.

My principal concern about the paper is the methodology followed by the authors; there is a lot of information missing about how the data for the study was obtained, the study period, and the methods. I highly recommend that the authors pay attention to the following major and minor observations:

Response: Thanks for your review. Based on your major and minor observations, we considered many revisions in the new version which can cover information missing in the manuscript. Please see our responses

Major observations

In the sections "Ecological degradation assessment (rivers)" and "Ecological degradation assessment (wetland)". The use of the fish species must be justified. How were the fish observations, flamingo observations, and expert opinions obtained? How many fish observations and expert opinions were used to develop the fuzzy functions? How many flamingo observations were used to develop the ANFIS model? Why is the abundance of flamingos an indicator of ecological degradation? For both models, how were the data partitioned to validate the models? Which is the DEM resolution?

Response: Based on your important comment. Two paragraphs are added in the new version. First one related to rivers and second one related to the wetland as follows. They are highlighted in the new version as well

More information regarding ecological modelling in river can be helpful for the readers. The target species was selected based on long-term fish observations related to several studies in this region and initial ecological studies in the study area. In fact, initial studies indicated that this species can be observed in all rivers in the wetland basins which means it can be a target species for all physical habitat simulations. Moreover, its population is highly sensitive to the physical flow parameters which means it can indicate the impact of changing physical habitat for assessing ecological flows. The fish observations were carried out by electrofishing method as one of the known methods for ecological observations in the river habitats. Then, results of the fish observations were discussed in an expert panel including two regional ecologist and the research team to convert the direct observations as well as other long-term observations to the fuzzy rules. The fish observations were carried out in more than 100 points in winter and summer of 2017 in which depth, velocity and bed particle size were measured simultaneously. In the hydraulic simulation a DEM file with 10 m resolution was applied. Furthermore, we defined three membership functions including low, medium, and high considering three physical parameters including depth, velocity, and channel index (bed particle size) which means 27 fuzzy rules as the expert opinions were developed for physical habitat simulation.

Many previous studies had been carried out for better understanding on the ecological challenges in the Urmia Lake (e.g., Sima et.al, 2021). All these studies highlighted that the flamingo is one of the terrestrial species which is highly dependent on the water level and total dissolved solids (TDS). In other words, changing either water level or TDS can change the Flamingo population considerably. Moreover, the Flamingo is the only major terrestrial species which would stay in the lake habitats for a long time during the immigration. Hence, all these environmental considerations justify selecting it as the environmental indicator for the wetland. Flamingo observations have been carried out by regional environmental department in the recent decade (in different years of the recent decade) which was applied in this study for developing a data driven model. Based on comparing results of the model and observations considering equations 1 and 2 as the evaluation indices, the Flamingo model was validated. The results will be presented in the result and discussion.

For the optimization model, How many hydrometric stations were used, and where are they located?

Response: Hydrometric stations were located at upstream and downstream of all rivers (8 main rivers as the inflow). In the study area, this issue has been mentioned.

It is important to mention the dates on which the data were obtained (flamingo abundances, crops, fish habitats, flows, and water depth) and analyzed (simulated period).

Response: Thanks for your constructive comment. Based on your previous comments, we added them in different parts. Please see highlighted parts in the new version for observing them.

Where are the fish in the results?

Response: It was a broad research work with a focus on cropping pattern which means showing all details of results is not possible in one manuscript. Hence, we tried to show the main results only. Figure 9 is the main outputs from fish habitat simulation. Displaying other details such as expert opinions in a long table and other details would prolong the manuscript unnecessarily.

The discussion has a lot of affirmations without support; please support them. Please consider writing about the spatial pattern of the crops inside the basin.

Response: I enhanced discussion based on your comment. Major changes are highlighted.

Minor Observations

Use the appropriate citation and reference styles for the journal.

Response: Revised

Line 123 specifies that the aquatic species is a fish.

Response: Revised

Line 124 Put the scientific name If you do not know the specific species, just put Phoenicopterus sp.

Response: Revised

Figure 1 and Figure 2 increase the size of each graph and put them on a horizontal

Response: The size is increased. As we checked, it is now clear for reading. Moreover, original excel file will be provided for the MDPI editor , if needed.

Figure 10: Sample No. of What? Is it a time associated with the sample No.?

Response: As mentioned, samples are the result of long-term studies (not a continuous sampling) and in each sample TDS and water level are known which means the time of the sampling cannot be mentioned in the axes and only the sample NO is needed in the axes.

Line 266, figure 8?? I think it's 11. WUA, not NWUA

Response: Revised

Figure 11 I do not understand why the NSE is observed vs. the model?

 Response: As mentioned in the manuscript, we used NSE for assessing the optimal instream flow as well. In figure 11, the natural flow is considered as the observations (ideal condition) and the optimal flow as the simulation (optimal condition) to investigate how natural flow and optimal instream flow are close in terms of WUA

Figures 11 and 12 Please use the same colors for the same variables.

Response: You can see the nature of these figures are not the same. Figure 11 is related to WUA (two time series) but figure 12 is related to flow (three time series)

Reviewer 2 Report

1. The research process and conclusion of the manuscript is unclear. It is lacked of the basic data for calculating, such as the area of wetland and the crops, .

2. The graphics were drawn rough, such as Fig. 2, Fig.3 Fig.13, Fig.14.

3. The optimization model and constraint conditions should be introduced in detail, and expressions should be provided.

4. Is the optimization model (Formula 3) reasonable? What is the relationship between the optimization model withimproving environmental water supply in the title?  How to calculate the DRV, DFC, and DWUA in the equation? What is the unit of DRV, DFC, and DWUA?

5. line 160: Why the fuzzy methods are advantageous in this study?

6. line 161: Why choosing the lower part of the river?

7. What is x in Figure 9?

8. What is the ' optimized cropping pattern' for the 'improving environmental water supply of wetlands' in the research area?

Author Response

  1. The research process and conclusion of the manuscript is unclear. It is lacked of the basic data for calculating, such as the area of wetland and the crops

Response: The selected crops are available. However, we added a table in which all these data are available

  1. The graphics were drawn rough, such as Fig. 2, Fig.3 Fig.13, Fig.14.

Response: The original excel files of the graphs will be sent to the MDPI editor for replacing in the proofreading, if the quality is not enough

The optimization model and constraint conditions should be introduced in detail, and expressions should be provided. Is the optimization model (Formula 3) reasonable? What is the relationship between the optimization model with“improving environmental water supply ”in the title?  How to calculate the DRV, DFC, and DWUA in the equation? What is the unit of DRV, DFC, and DWUA?

Response: Based on your comment, we added a paragraph in the optimization section as the response to all these questions. Highlighted in the new version

  1. line 160: Why the fuzzy methods are advantageous in this study?

Response: We added a paragraph in this regard in the discussion because it was related to the discussion, and we explained why fuzzy method is advantageous in this study. Please see highlighted texts in the discussion

  1. line 161: Why choosing the lower part of the river?

Response: We added the following text to the manuscript as the response of your question

If enough environmental flow can be available at downstream of the rivers, it will mean all fish habitats from upstream to the downstream would have adequate instream flow. Moreover, available flow at downstream (after the selected river reaches) of the rivers will drain to the lake because no more water abstraction is available at downstream

  1. What is x in Figure 9?

Response: It was a minor mistake which is corrected in the new version. It is inflow as well

  1. What is the ' optimized cropping pattern' for the 'improving environmental water supply of wetlands' in the research area

Response: Figure 15 shows these optimal cropping pattern for each year of the simulation period which will improve supply of environmental water requirement

Reviewer 3 Report

This article is on the topic of improving environmental water supply of wetlands through optimal cropping pattern. The research work is technically strong and I have a few minor comments:

1.     Novelty statement: the authors elaborated the limitations of the existing studies in the first two sentences in the last paragraph of Introduction. More in-depth discussions are needed to explain the research gap. On the other hand, the authors can trim the discussions above in the Introduction, leaving more space to discuss the gap and how they address the research gap.

2.     There are 15 figures in this manuscript. Many of them can be placed into Supporting file. Figure 14 and 15 have overlapped figures.

Author Response

This article is on the topic of improving environmental water supply of wetlands through optimal cropping pattern. The research work is technically strong and I have a few minor comments:

  1. Novelty statement: the authors elaborated the limitations of the existing studies in the first two sentences in the last paragraph of Introduction. More in-depth discussions are needed to explain the research gap. On the other hand, the authors can trim the discussions above in the Introduction, leaving more space to discuss the gap and how they address the research gap.

Response: Thanks for your comment. We embellished the discussion by adding text regarding why this research gap is important and how we addressed it. Please see highlighted text (discussion) in the new version

  1. There are 15 figures in this manuscript. Many of them can be placed into Supporting file. Figure 14 and 15 have overlapped figures.

Response: Please note that the nature of these two figures are different although some limited overlaps can be observed such as total cultivated area. However, figure 15 highlighted the details of cropping pattern and figure 14 shows net revenue and total area and other environmental parameters. Moreover, these figures are key outputs which should be part of the paper to inform the readers regarding how the key outputs should be shown

Round 2

Reviewer 1 Report

Accept in present form

Reviewer 2 Report

The author did not make comprehensive revisions to the existing problems.

Most graphics are not clear.